# The Effect of Identity Salience on Residents' Engagement with Place Branding during and Post COVID-19 Pandemic

**Wei Han, Yuwei Tang and Jiayu Wang *** 

Tourism School, Hainan University, Haikou 570228, China
* Correspondence: jiayu.wang@hainanu.edu.cn

**Abstract:** It is critical for tourism managers and marketers to understand how to support the sustainable recovery of the industry by involving residents in tourism marketing. The purpose of this study is to investigate whether the COVID-19 pandemic, as an environmental stimulus, could enhance the salience of residents' collective identity, thereby increasing their engagement with place branding. A conceptual framework was developed to explore the role of relationship-inducing factors and non-relationship-inducing factors in activating residents' collective identity based on the social identity salience theory of relationship marketing success. The research model was tested using data from 603 questionnaires collected from Sanya, China, during the period of a sudden COVID-19-related crisis. The findings from the statistical analyses showed significant relationships among the research variables, and the moderating role of perceived social connectedness was supported. Our findings have implications regarding how to engage residents in the tourism recovery process during and post the COVID-19 pandemic.

**Keywords:** identity salience; COVID-19; resident engagement; social connectedness

## 1. Introduction

The COVID-19 pandemic has intensified the tension between tourists and residents, with extensive evidence reporting residents' conflicts and discontentment with tourists (Tung et al., 2021; Ryu et al., 2022) [1,2]. In the "new normal" and post-pandemic era, it is critical for tourism managers and marketers to understand how to support the sustainable recovery of the industry by involving residents in tourism marketing. Indeed, residents have been dynamic agents in overcoming the negative consequences of the pandemic. For example, studies have found that the information disseminated by residents could attract more public attention than that of tourism institutions during the pandemic [3,4].

Meanwhile, the pandemic has increased the sense of shared identity amongst residents within communities [5]. Studies focusing on posttraumatic growth suggest that the pandemic has led to greater group strengths and enhanced group identity [6,7], which facilitates social support and enhances a sense of control [8]. Due to the psychological proximity between the residents and their communities, the COVID-19 pandemic has changed public behavior by shaping social norms and identities [5]. To gain more social resources, individuals tend to enhance their self-evaluation and form desired social identities [2,9,10]. As a stimulus cue, the COVID-19 pandemic could make a particular identity salient. For example, Neville et al. (2021) reported that, during the pandemic, when the Scottish identity was salient, residents were more willing to wear masks to protect community members [5].

Hence, during the pandemic, facing a common fate has increased the sense of shared identity amongst residents within communities, which has increased collective resilience and acts of solidary [5,11]. This may provide an opportunity for local institutions and destination managers to engage residents in tourism support activities in a post-pandemic era, as well as to make the industry more sustainable in the future. Previous studies have highlighted residents' key role in destination marketing [12–14], and particularly, the

contribution of residents to destination branding by acting as brand ambassadors [14–16]. However, it is unclear whether the sense of "common fate" caused by the pandemic could impact residents' attitudes and behavior toward tourism. To our knowledge, at present, limited studies have investigated how the COVID-19 pandemic has impacted residents' engagement with destination branding by shaping their social roles and identity.

The purpose of this study is to investigate whether the COVID-19 pandemic, as an environmental stimulus, could make residents' collective identity salient, thereby increasing their involvement and participation in destination branding activities. Specifically, by applying the identity salience theory, we explore the role of relationship-inducing and non-relationship-inducing factors in activating residents' collective identity and enhancing residents' engagement in destination branding. Further, to provide a more nuanced understanding, we investigate the moderating role of social connectedness, as a cue of motivation for regular and positive interaction with others [17].

Accordingly, we developed a conceptual framework and collected data from residents of a tourism-dependent city in China to investigate whether residents were willing to engage in destination-branding activities when a COVID-19-related crisis occurred. In doing so, our findings can help to determine the complex mechanisms by which residents engage in tourism branding. Realizing residents have been a dynamic agent in overcoming the negative consequences of the pandemic, this work will enrich the extant literature on sustainable tourism and destination marketing, especially during the unprecedented times of the COVID-19 pandemic. Practically, our findings will help local institutions and destination managers to understand how to engage residents in destination branding and developing favorable host-guest relations by fulfilling their social roles.

## 2. Literature Review and Hypothesis Development

### 2.1. Identity Salience and Residents' Engagement with Place Branding

The concept of identity salience is driven from the social identity theory, which posits that identities are the consequence of the process of self-categorization through which individuals categorize themselves on different levels of abstraction [18], and each individual processes many different identities over time [19,20]. Indeed, identity is fluid and impacted by contextual cures, rather than being unchanging and stable [21]. Identity salience, defined as "the relative importance or centrality of given identity (role) for defining oneself" [22], impacts individuals' choices of certain activities. When an individual perceives a given situation as an opportunity for performing activities related to a certain identity that has high salience, they may perform those activities in accordance with that identity [23,24]. Studies of identity salience focus on the extent to which a specific social identity becomes maximally meaningful in a certain context [21,25]. For instance, Stephan et al. (2009) argued that a collective identity can become salient because of a perceived threat and exclusion. Mastro and Atwell (2012) and Reid et al. (2012) both found that certain conversation topics and media exposure can make a particular identity salient (e.g., national identity) [26,27]. Xu (2020) found that social and cultural issues, such as an environmental crisis, affect employees' identity salience [21]. The contexts in which individuals check and reflect on their privileged identities are called "identity cues" or "stimulus cues" [21,28].

During interactions, individuals tend to centralize the situationally salient identities because these identities impact how they perceive their sense of connection with others [21, 25]. As a result, identity salience theory has been widely applied in relationship marketing and public relations studies [29–31]. It has been found that identity salience can satisfy individuals' self-definition and self-enhancement needs and further prompt actions as an expression of the salient identity. For example, Arnett et al. (2003) evidenced that identity salience significantly impacts alumni's donation behavior [29]. Additionally, Michalski and Helmig (2008) revealed a link between identity salience and employees' organizational support behaviors, such as positive word-of-mouth behavior and donations [32]. Xu (2020) found identity salience can make individuals perceive the organization as "one of us", thus positively influencing their identity expression behavior [21].

In the tourism field, place branding involves many stakeholders and constant stakeholder interaction keeps a place brand active and in constant formation [3]. Any negative events regarding these stakeholders would profoundly impact the sustainable development of tourism destinations, leading to negative tourist reactions, such as perceptions of betrayal and boycott [33,34]. Studies have revealed a direct relationship between identity salience and residents' tourism related behavior. For example, Lee et al. (2016) reported that identity salience impacts tourists' leisure pursuit behavior [23]. Simpson and Siguaw (2008) showed that, when a place is perceived as being identity salient, the residents and tourists promote the place by word-of-mouth [35]. Furthermore, according to Wassler et al. (2019, 2021), place branding refers to the effort made by tourism-oriented groups and authorities to market the place as a tourist site, which is closely related to social identity [14,16]. Place constructs residents' identity and on the other hand, as place ambassadors and authentic narrators of place culture, residents are significant dynamic interpretants of place image [36]. Residents who identify themselves with the region as the whole of discourses, institutionalized practices and symbols would actively participate in place branding [37]. Therefore, we propose that there is a direct relationship between residents' identity salience and residents' engagement with place branding:

**H1.** *A positive relationship exists between residents' identity salience and residents' engagement with place branding.*

### 2.2. Relationship-Inducing and Non-Relationship-Inducing Factors

Arnett et al. (2003) developed "the identity salience model" to explain the underlying role of identity salience in the success of relationship marketing [29]. They found that both relationship-inducing factors and non-relationship-inducing factors can impact relationship marketing success. Firstly, they argued that identity salience is especially critical in contexts that facilitate the exchange of substantial social benefits among partners, which is impacted by the quality and number of social exchanges related to the identity. When relationship-inducing factors support or confirm the identity, identity salience may occur. Specifically, Arnett et al. (2003) found that the relationship-inducing factors of participation, reciprocity, prestige, and satisfaction impact alumni's donation behavior. Furthermore, they found that other non-relationship-inducing factors, such as economic benefits and perceived need, also impact people's behavior [29].

Although identity salience theory has been applied to investigate relationship marketing success [23,29,38,39], a model focusing on the antecedents of identity salience has not been considered in explaining residents' pro-tourism behavior, especially not in the context of the COVID-19 pandemic. The pandemic, as an unprecedented and ongoing crisis, has led to significant social change [5,40,41]. By considering identity salience theory, we developed a conceptual framework to examine the effect of relationship-inducing factors and non-relationship-inducing factors in explaining residents' engagement with place branding. Specifically, in this study, we use community cohesion, psychological empowerment and satisfaction with government as three relationship-inducing factors because they are critical factors that can facilitate identity-related social relationship exchange and social interactions during and post the COVID-19 pandemic [7,42–44]. In contrast, we include economic benefits and perceived risk as two non-relationship-inducing factors because they can reflect the trade-off faced by the residents in a post-pandemic era, and both have significant effects on residents' tourism-related attitudes and behavior [20,42,44–46].

#### 2.2.1. Relationship-Inducing Factors and Identity Salience
Community Cohesion

The COVID-19 pandemic has changed personal social networks and social roles [7,41]. Additionally, the pandemic has enabled individuals to develop stronger social relationships with their close neighbors, thus increasing community cohesion, as community members have received more social support from their neighbors [7,43]. From the tourism per-

spective, Hajibaba et al. (2017) argue that crises help involve residents in the response to long-term processes: they found that during a crisis, residents are more willing to share available resources with tourists as a form of support [47]. Additionally, after a crisis, communities are characterized by greater levels of trust, cohesion, and civic engagement [48]. Miao et al. (2022) highlighted that the tourism industry has benefited from posttraumatic growth to build stronger interpersonal relationships [49].

Community cohesion refers to the sense of bonding and togetherness exhibited by community members [50]. As an indicator of intergroup connection, it can represent social identity [51], especially during and after a crisis. For example, Chang (2010) found that community cohesion increased as residents increasingly realized the significance of community unity and came together to cope with the losses experienced after a flood [52]. Wetherell (2007) argued that, when a crisis occurs, community cohesion encourages individuals to identify with super-ordinate identities such as "the whole neighborhood" and "the community in general" [53]. Based on the above discussion, we propose the following hypothesis:

**H2a.** *A positive relationship exists between community cohesion and identity salience.*

Psychological Empowerment

Psychological empowerment is defined as "the increased intrinsic task motivation manifested in cognitions that reflect an individual's active orientation to his or her work role" [54]. As argued by Wassler et al. (2021), although communities are highly heterogeneous, the struggle for empowerment makes residents behave homogeneously [16]. Previous studies suggest that psychological empowerment impacts residents' perceptions and behavior toward tourism [16,42,55]. For example, Koelble (2011) found that psychological empowerment makes residents feel they can represent the place, which contributes to the development of sustainable tourism and ecotourism [55]. Additionally, Wassler et al. (2021) revealed a link between psychological empowerment and residents' brand ambassadorship behavior [16]. Finally, Li et al. (2022) reported that the effect of psychological empowerment on residents' support for tourism became more prominent during the COVID-19 pandemic when gaming tourism was affected [42].

Social identity theory suggests that empowerment can facilitate collective action by fueling social change [56]. Indeed, psychological empowerment can increase individuals' confidence in collective actions, and help individuals develop an empowered self-concept as a consequence of role transformation from the out-group to the in-group [57]. Empowered individuals tend to feel that their roles fit their personal values, believe in their own capabilities, have a sense of autonomy, and, ultimately, believe they can impact the outcome of the process [58]. Therefore, empowerment has been frequently used to study identity-related collective behavior. For example, Rawat (2011) demonstrated that employees' psychological empowerment can make them behave on behalf of the organization to reinforce their role within the organization [59]. Additionally, Logan and Ganster (2007) found that psychologically empowered employees view themselves as "co-owners" of an organization, which shapes their perceived responsibility toward the organization [60]. In the context of the COVID-19 pandemic, psychological empowerment may induce a sense of unity with the community and help activate individuals' collective identity, thus leading to the following hypothesis:

**H2b.** *A positive relationship exists between psychological empowerment and identity salience.*

Satisfaction with the Government

According to Arnett et al.'s (2013) identity salience model, satisfaction is an important antecedent of identity salience. Additionally, the significance of satisfaction on identity salience has been verified by several studies in the organizational context [29]. For example, Welborne and Cable (1995) showed satisfaction with the organization helped people reevaluate the salience of different identities [61]. McCall and Simmons (1978) argue satisfaction with organizational events was significant for the development and maintenance of employees' identities [62]. In the tourism field, as an indicator of residence-place relationship,

satisfaction with the local government has a direct impact on residents' social identity and identification with the place [63,64]. Studies have revealed that satisfaction with the local government is positively related to residents' perceived positive impact on the local community and tourism activities. For example, Wassler et al. (2021) found that trust in the government impacted residents' identification with the place and ambassadorial behavior [16]. Zuo et al. (2017) reported that mainland Chinese residents showed a higher level of support for "red tourism" if they were satisfied with the local government [65].

During the COVID-19 pandemic, the development of the travel industry has been highly contingent on local government travel restrictions [66], meaning residents' attitudes and behavior toward tourism have been directly impacted by government policies [44]. The significance of government in tourism post-pandemic recovery has been revealed by several studies. For example, Park et al. (2021) found residents' satisfaction with government COVID-19 prevention measures significantly impacted their attitude toward tourism [67]. Wong and Lai (2021) argue during the COVID-19 pandemic, satisfaction with the government can increase residents' confidence and support for tourism recovery [68]. Furthermore, Fakfare and Sangpikul (2022) found residents' satisfaction with the government impacted their perception of tourism reactivation in a post-pandemic era [69]. Therefore, we posit that satisfaction with the government can impact residents' identity salience:

**H2c.** *A positive relationship exists between satisfaction with the government and identity salience.*

2.2.2. Non-Relationship-Inducing Factors and Residents' Engagement with Place Branding
Perceived Economic Benefits

Extensive evidence shows that the economic benefits of tourism play a critical role in residents' engagement and pro-tourism activities [44,70–72]. Indeed, by applying the social exchange theory, scholars have demonstrated a direct relationship between perceived economic benefits and residents' pro-tourism behavior [44,73,74]. For example, studies have identified connections between the individual economic benefits gained from tourism and a wide range of tourism-related behaviors, such as positive word-of-mouth behavior [64], brand ambassadorial and citizenship behavior [75], and support for "red tourism" [65].

COVID-19-related studies perceived economic benefits from tourism are a key driver of residents' pro-tourism behavior. For example, Li et al. (2022) suggested that the perceived economic benefits impacted residents' support for gaming tourism when the pandemic broke out [42], and Woosnam et al. (2022) revealed a direct positive relationship between perceived economic benefits and residents' pro-tourism behavior [44]. Hence, we propose the following hypothesis:

**H3a.** *A positive relationship exists between perceived economic benefits and residents' engagement with place branding.*

Perceived Risk

Perceived risk refers to people's assessment and awareness of the negative consequences that may result from their decisions and reflects individuals' expectations of the potential loss [46,76]. Individuals tend to use risk-handling strategies to avoid loss, even when there are considerable potential benefits from risk-taking [77]. Due to the potential of the pandemic to incur substantial social costs to local communities, residents' perceived risk may hurt their attitudes and behavior toward tourism [46,78].

Previous studies have examined the negative relationship between the perceived risk of COVID-19 and residents' pro-tourism behavior [44,45,78]. For example, Woosnam et al. (2022) reported a direct negative relationship between the perceived risk of COVID-19 and residents' attitudes and behavior toward tourism [44]. Additionally, Armutlu et al. (2021) showed that Turkish people's perceived risk was negatively related to their hospitality behavior [45]. Furthermore, Ryu et al. (2022) demonstrated that the risk perceptions associated with tourism made residents unwilling to accept international tourists because

of self-protection motivations during the pandemic [1]. Based on the above findings, we develop the following hypothesis:

**H3b.** *A negative relationship exists between perceived risk and residents' engagement with place branding.*

*2.3. The Moderating Effect of Social Connectedness*

Social connectedness, which gives an internal sense of belonging, means the perceptional awareness of the close emotional relationship between the self and others [79,80]. Social connectedness is of great importance in the context of the COVID-19 pandemic because the pandemic has greatly changed individuals' social relationships and perceived connectedness with others [7]. Measures such as community lockdowns, social distancing, and travel restrictions have increased individuals' reliance on other community members for connection and closeness [6]. Importantly, social connectedness enables individuals to better prepare for, respond to, and recover from disasters [81].

Specifically, during crises, social connectedness plays a critical role in activating individuals' identity systems and encouraging them to behave in line with their social roles. Studies have reported that social connectedness impacts individuals' perception and behavior toward in-group members [82–84]. For example, Pitas and Ehmer (2020) found that social connectedness with immediate household members and the local community impacts people's collective response to COVID-19 [82]. Zetterberg et al. (2021) showed that social connectedness positively influences community social interactions and support during the epidemic [84].

Overall, social connectedness is an important inner concept that strengthens the effects of individuals' social roles [85,86], because it elicits a sense of self-worth in the group and enhances the desire for additional connection with others [17]. Indeed, Gieling and Ong (2016) demonstrated that social connectedness enables individuals to identify strongly with their group members and drives identity-enhancement motivation [87]. Additionally, a study by Grewal et al. (2019) showed that stronger social connectedness increases individuals' engagement in social media information-sharing for social and normative needs [88]. Based on the above discussion, we present the following hypothesis:

**H4.** *Perceived social connectedness moderates the relationship between identity salience and residents' engagement with place branding in that the relationship is stronger for those who have higher perceived social connectedness.*

Figure 1 illustrates our proposed hypotheses.

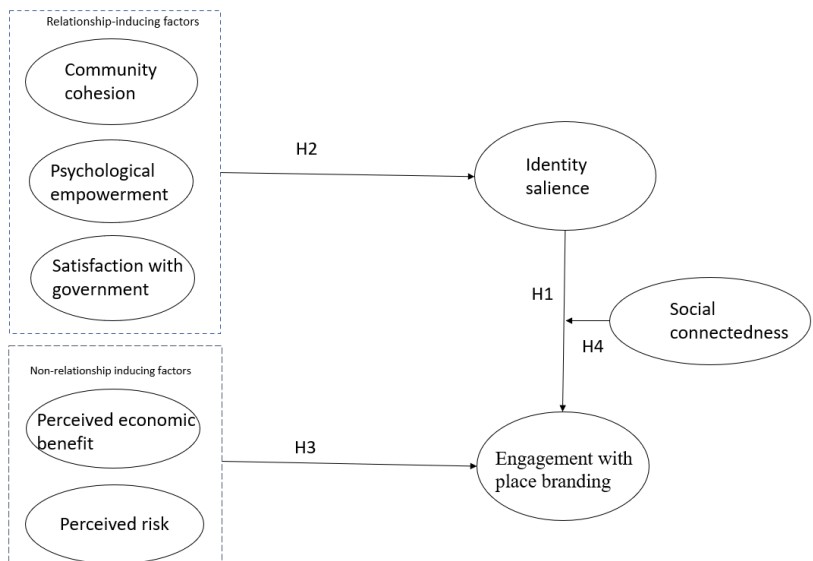

**Figure 1.** Proposed hypotheses.

## 3. Research Method

### 3.1. Participants

The target population for this study was adult residents of Sanya city, Hainan Province, China. Sanya is famous for its beach resorts and tropical climate. Since 1998, the Sanya government has identified tourism as its priority development strategy. As one of China's most famous tourism destinations, the economic revenue of the Sanya government relies heavily on tourism. During the pandemic, the strict overseas travel restrictions enacted by the Chinese government led to an increase in domestic travel. In 2021, the revenue generated from the tourism sector was 74.7 billion, accounting for over 60% of Sanya's GDP [89]. The data for this study were collected between 10 August 2022, and 25 August 2022, when Sanya's tourism industry was in crisis. At this time, the government enacted a strict travel restriction because of the sudden increase in the number of COVID-19 infections. On 6 August 2022, more than 80,000 tourists were required to quarantine for at least 14 days before leaving Sanya, leading to a heated discussion on the internet. Although the Sanya government had taken measures to ameliorate these negative responses, such as by providing discounted accommodation, Sanya's tourism industry suffered greatly.

To understand the residents' reactions and their perceptions toward the tourism industry amidst the crisis, we accessed the Sanya residents through Wenjuan.com and designed two filter questions to ensure all the respondents were Sanya residents aged 18 years or older. We obtained 630 useful questionnaires from respondents. Table 1 shows respondents' demographic characteristics.

**Table 1.** Sample characteristics (N = 630).

| Demographic Characteristics | Frequency | Percentage |
|---|---|---|
| **Gender** | | |
| Female | 283 | 44.9 |
| Male | 347 | 55.1 |
| **Age** | | |
| 18–30 | 211 | 33.5 |
| 31–40 | 174 | 27.6 |
| 41–50 | 82 | 13.0 |
| 51+ | 163 | 25.9 |
| **Household income (per month)** | | |
| Less than USD 719 | 127 | 20.1 |
| USD 719-USD 2158 | 342 | 54.3 |
| More than USD 2158 | 161 | 25.6 |
| **Education** | | |
| Middle school or lower | 57 | 9.1 |
| High school/vocational school | 162 | 25.7 |
| College/university | 317 | 50.3 |
| Master's or Ph. D. | 94 | 14.9 |

### 3.2. Measures

All the items employed to measure constructs within the conceptual model were adapted from existing measures employed in previous studies. We made appropriate wording changes to the items to ensure they fit our study context. We used four items adapted from Arnett et al. (2003) to measure identity salience [29]; six items adapted from Chen and Dwyer (2018) to measure residents' engagement with place branding [63]; six items adapted from Somer et al. (2008) to measure community cohesion [90]; five items adapted from Li et al. (2022) to

measure psychological empowerment [42]; three items adapted from Wong et al. (2022) to measure satisfaction with the government [91]; four items adapted from Boley et al. (2018) to measure the perceived economic benefit from tourism [73]; four items adapted from Woosnam et al. (2021) to measure the perceived risks from COVID-19 [44]; and three items adapted from Kim et al. (2022) to measure perceived social connectedness [86]. These measurement items were adopted from previous studies demonstrating high reliability and validity and are suitable to be used in a time of COVID-19. All items were measured using a seven-point Likert scale ranging from 1 = totally disagree to 7 = totally agree. Then, we conducted a pretest and 11 items were eliminated due to low communality or cross-loading of more than one factor. Finally, a questionnaire with 24 measurement items were generated (Appendix A).

*3.3. Procedure*

We performed initial data screening and the common method bias test to confirm the data were suitable for further analysis. Firstly, we calculated the VIF values to test the collinearity. All the VIF values for the constructs in this study were less than 2, indicating the absence of multicollinearity issues [92]. Secondly, we conducted skewness and kurtosis tests to ensure the data were normally distributed [93]. Thirdly, Harman's one-factor test was used to examine common method bias. The results revealed that no single factor accounted for more than 40% of the variance, providing evidence of no common method issues within the measures. Fourthly, we used control variables, including gender, age, household income and working industry, to ensure the observed effects were attributed to the core constructs. This helps control the effects of possible confounders, thereby improving internal validity [94].

The research model was examined by PLS-SEM with Smartpls 3.3 software. PLS has paramount predictive accuracy, which is suitable for developing theory [95]. For this study, we aim to explore the role of relationship-inducing factors and non-relationship-inducing factors on residents' engagement with destination branding, thus the PLS model was preferred.

## 4. Results

*4.1. Confirmatory Factor Analysis (CFA)*

The reliability and validity of the measurement model were evaluated. First, the factor loadings of all the indicators exceeded the minimum criteria of 0.5, and the average variance extracted (AVE) from each construct exceeded the threshold of 0.5, supporting satisfactory convergent validity. Second, the composite reliability coefficients were greater than 0.7, indicating good reliability [93] (Table 2). Third, the discriminant validity was verified with Fornell and Larcker's (1981) criteria [96]. As shown in Table 3, all the intercorrelations were no larger than the square root of the AVE of the constructs, verifying all the constructs were different.

**Table 2.** Item loadings and validities.

| Scale and Item Description | Loading | Cronbach's Alpha | AVE | Compositive Reliability |
|---|---|---|---|---|
| **Identity salience (IS)** | | | | |
| Sanya is a significant part of me. | 0.93 | | | |
| For me, Sanya means more than where I am living. | 0.93 | 0.93 | 0.88 | 0.95 |
| The image of Sanya is something I usually think about. | 0.94 | | | |
| **Residents' engagement in place branding (RE)** | | | | |
| I contribute to improving residents' experience in Sanya. | 0.91 | | | |
| I make efforts to strengthen the image of Sanya as a tourism destination. | 0.94 | 0.92 | 0.87 | 0.95 |
| I "talk up" Sanya to people I know. | 0.95 | | | |

**Table 2.** *Cont.*

| Scale and Item Description | Loading | Cronbach's Alpha | AVE | Compositive Reliability |
|---|---|---|---|---|
| **Community cohesion (CC)** | | | | |
| I am closely related to my community. | 0.92 | | | |
| I have developed fulfilling relationships with other members of the community. | 0.86 | 0.87 | 0.80 | 0.92 |
| I am always willing to help others in the community. | 0.92 | | | |
| **Psychological empowerment (PE)** | | | | |
| Tourism in Sanya makes me proud to be a Sanya resident. | 0.90 | | | |
| Tourism in Sanya makes me feel special because my city is famous. | 0.86 | 0.88 | 0.81 | 0.93 |
| Tourism in Sanya makes me want to talk more about my city. | 0.93 | | | |
| **Satisfaction with government (SWG)** | | | | |
| The government made the right tourism decisions when the pandemic broke out. | 0.94 | | | |
| The government did what was right in tourism when the pandemic broke out. | 0.94 | 0.94 | 0.90 | 0.96 |
| The government looked after the community's interests during the pandemic. | 0.96 | | | |
| **Perceived economic benefit (PEB)** | | | | |
| My family income is tied to tourism. | 0.95 | | | |
| My income has decreased sharply because of the Sanya tourism industry's suffering. | 0.94 | 0.92 | 0.87 | 0.95 |
| My economic future depends on tourism. | 0.96 | | | |
| **Perceived risks (PR)** | | | | |
| Tourists make me feel anxious during the pandemic period. | 0.80 | | | |
| Tourists increase the risk of COVID-19 infection. | 0.81 | 0.66 | 0.60 | 0.81 |
| Tourists increase inconveniences in my daily activities. | 0.68 | | | |
| **Perceived social connectedness (SC)** | | | | |
| I feel closely connected with my community during the pandemic. | 0.91 | | | |
| I feel positive toward people connected with me during the pandemic. | 0.94 | 0.88 | 0.81 | 0.93 |
| Connection with others increases my belongingness during the pandemic. | 0.95 | | | |

**Table 3.** Discriminant validity.

|  | CC | IS | PE | PEB | PR | RE | SC | SWG |
|---|---|---|---|---|---|---|---|---|
| **CC** | **0.89** | | | | | | | |
| **IS** | 0.21 | **0.94** | | | | | | |
| **PE** | 0.28 | 0.25 | **0.90** | | | | | |
| **PEB** | 0.21 | 0.81 | 0.24 | **0.93** | | | | |
| **PR** | −0.36 | −0.16 | −0.29 | −0.14 | **0.77** | | | |
| **RE** | 0.10 | 0.57 | 0.15 | 0.56 | −0.07 | **0.93** | | |
| **SC** | 0.34 | 0.18 | 0.27 | 0.18 | −0.35 | 0.10 | **0.90** | |
| **SWG** | 0.13 | 0.58 | 0.15 | 0.57 | −0.09 | 0.86 | 0.10 | **0.95** |

*4.2. Structural Equation Model Testing*

After validating the measurement model, the adjusted $R^2$ coefficient was used to assess the fit of the structural model. $R^2$ value for identity salience was 0.36 and for residents' engagement was 0.35, indicating the model explained 36% of the variance on identity salience and 35% of the variance on residents' engagement in destination branding. Additionally, a $Q^2$ test was performed to assess the predictive validity of exogenous latent variances [93]. The $Q^2$ value for identity salience was 0.32 and for residents' engagement was 0.30, indicating the model can predict identity salience and residents' engagement with place branding.

A non-parametric bootstrapping procedure—with 603 cases, 5000 subsamples, and individual sig changes—was used to test research hypotheses [93]. As shown in Table 4 and Figure A1, Hypotheses 1–4 were all supported. Both relationship-inducing factors and non-relationship-inducing factors are significantly related to residents' engagement in destination branding. Perceived social connectedness had a significant moderating effect on the relationship between identity salience and residents' engagement in destination branding.

**Table 4.** Results of Hypothesis testing.

| Hypothesis | Path Coefficients | *T* Values | *p*-Values | Decision |
|---|---|---|---|---|
| H1: Identity salience-> Resident engagement | 0.31 | 2.99 | <0.01 *** | Supported |
| H2a: Community cohesion-> Identity salience | 0.10 | 4.11 | <0.01 *** | Supported |
| H2b: Psychological empowerment-> Identity salience | 0.14 | 5.54 | <0.01 *** | Supported |
| H2c: Satisfaction with government-> Identity salience | 0.55 | 16.00 | <0.01 *** | Supported |
| H3a: Perceived economic benefits-> Resident engagement | 0.27 | 2.58 | <0.01 *** | Supported |
| H3b: Perceived risk-> Resident engagement | −0.10 | 2.60 | <0.01 *** | Supported |
| **Moderating effect** | | | | |
| H4: Perceived social connectedness × identity salience -> Resident engagement | 0.15 | 5.01 | <0.01 *** | Supported |

*p* < 0.01 ***

**5. Discussion and Conclusions**

The COVID-19 pandemic has led to serious negative social and psychological consequences for tourism stakeholders. As a result, there is increased research interest in the resilience of the tourism industry, which is trying to capitalize on opportunities that have arisen due to the crisis [40,97,98]. Several studies have highlighted the importance of residents in the period of tourism recovery [44,46,91], but few of them focused on residents' active engagement with place branding. To address this research gap, this study constructed a conceptual model that applied the identity salience theory to examine the role of relationship-inducing factors and non-relationship-inducing factors in residents' engagement with place branding. This study has implications regarding how to better engage residents in place branding by activating their collective identity during and after a crisis.

Focusing on residents' engagement in destination branding during and after the pandemic, our findings highlight the significance of identity salience. Although studies suggest that collective trauma can induce personal growth, strengthened group ties, and more collective social benefits in the post-pandemic period [99,100], few of them explored how these changes can impact individuals' social roles and collective behavior. Considering the significant role of social identity in protecting members from stressful situations and coping with crises [11], our study employed identity salience as a dominant predictor of residents' engagement in destination branding. The findings indicate that identity salience can positively impact residents' engagement in destination branding amid the COVID-19 pandemic, with a path coefficient of 0.35. This is in line with the results of other studies suggesting that facing a common crisis can encourage people to act in solidarity by highlighting their shared identity [81].

Using community cohesion, psychological empowerment, and satisfaction with the government as three antecedents of identity salience, we examined the relationship between relationship-inducing factors and identity salience. Although studies suggest that these factors can evoke collective identity and foster the development of a closer relationship with others [7,42,43], at present, none of them have explored their role in enhancing self-evaluation and activating desired social identities amid the COVID-19 pandemic. Our findings demonstrated significant positive relationships between these relationship-inducing factors and identity salience, thus suggesting that the COVID-19 pandemic, as an identity-relevant stimulus, can increase the salience of collective social identity through these relationship-inducing factors.

Additionally, we explored the effect of two significant non-relationship-inducing factors related to the COVID-19 pandemic: perceived economic benefits from tourism and perceived risk on residents' engagement with place branding. Our results show that both perceived economic benefits and perceived risk were found to have significant impacts on residents' engagement with place branding. These findings are in accordance with previous studies [42,44,46] conducted during the COVID-19 pandemic. The current empirical study contributes new evidence regarding the significance of perceived economic benefits and perceived risk on residents' attitudes and behavior toward tourism amid and after the COVID-19 pandemic [44,78,101].

The moderating effect of social connectedness on the relationship between identity salience and residents' engagement with place branding was examined. During the period of the COVID-19 pandemic, human behavior was greatly impacted by physical isolation caused by measures such as social distancing, community lockdowns, and travel restrictions [6]. In the "new normal" era of the pandemic, people may actively seek more connection, closeness, and belonging. To provide a more nuanced understanding of post-pandemic recovery, we tested whether perceived social connectedness moderates the relationship between identity salience and residents' engagement with place branding. Our findings supported a positive moderating effect of perceived social connectedness on this relationship. This finding is in line with previous studies that argue that social connectedness is an important inner concept that strengthens the effects of individuals' social roles and social identity [85,86]. Indeed, when individuals perceive a closer relationship with their group members, they increasingly engage in collective behaviors to support their salient social roles.

### 5.1. Theoretical Implications

The current research enriches the existing literature from various perspectives. Theoretically, we provide the first attempt at applying the identity salience theory to the context of tourism. Although scholars have applied frameworks—such as social support theory, social exchange theory, identity theory, and affect theory to examine travel behavior in the context of the COVID-19 pandemic [44,46,101]—residents' emotions and psychological well-being has been relatively overlooked. Furthermore, although the concept of identity salience has been widely employed by marketing and public relation studies [29,102,103], its application in the tourism field is limited. Focusing on this research gap, we proposed an integrated framework to explore the effect of identity salience on residents' engagement

with place branding, applying the identity salience theory. The findings demonstrated that identity salience, as a consequence of relationship-inducing factors, positively impacts residents' engagement in place branding.

Further, we integrated two non-relationship-inducing factors (perceived economic benefit and perceived risk) into our conceptual model. Our conceptual model explained 35% of the variance of residents' engagement in place branding, thus supporting the tenet of applying identity salience to explain residents' pro-tourism behavior. The relationship-inducing factors explained 36% of the variance of identity salience, supporting the idea that community cohesion, psychological empowerment, and satisfaction with the government are significant antecedents of identity salience in the context of the COVID-19 pandemic.

Finally, we investigated the moderating effect of perceived social connectedness on the identity salience model. One of the key opportunities for post-traumatic growth is the development of closer international relationships among key tourism stakeholders [43,49]. Although scholars argue that post-traumatic tourism development should focus on social relationships, ethics, and collective well-being [103,104], few studies have explored social connectedness as a motivation for regular and positive interaction with others amid the pandemic. By supporting the positive moderating effect of perceived social connectedness on the relationship between identity salience and residents' engagement in place branding, our results enrich the tourism crisis literature by highlighting the significance of social connectedness in engaging residents in tourism development after a crisis.

*5.2. Practical Implications*

Residents are faced with a dilemma between accepting tourists to avoid economic losses and intensifying conflicts and discontent with tourists [1,101]. Therefore, it is critical for destination managers and governments to consider how to encourage residents to support future tourism development. Our study focused on the effect of the pandemic on shaping residents' social relationships and social roles, and the findings provide practical implications from three perspectives.

Firstly, communities should foster favorable interpersonal relationships and highlight collective well-being, thereby increasing collective participation and collective efficacy. Specifically, from the perspective of community cohesion, attention should be given to the development of more integrated social networks for people to transfer, convert, and reinforce social capital. Local government and community managers can highlight the community as a unit and encourage residents to provide immediate support to each other. To increase psychological empowerment, community managers can emphasize the meaning of each individual with respect to the local network and value everyone's contributions to the development of collective efficiency and collective well-being. Since satisfaction with the government is of great importance to identity salience, the government is suggested developing appropriate strategies in responding to the crisis related to the COVID-19 pandemic and monitoring their implementation consistently. When a crisis occurs, the government should respond promptly to avoid losing public confidence. By taking these measures, individuals' collective identity can be activated, which should increase their willingness to actively engage in tourism development.

Secondly, this study highlights the importance of perceived risks and perceived economic benefits in relation to residents' engagement with place branding, which provides recommendations regarding the communication strategies of local government and tourism organizations. Indeed, efforts could be made to disseminate information about the economic benefits of tourism development to secure local support. Meanwhile, relevant stakeholders can increase the public's confidence by mitigating the perceived risk of tourism. For example, strategies that enhance the feeling of empowerment can also mitigate concerns about the risks related to COVID-19. Local governments can also highlight the strategies they have applied to reinforce the safety of the local community to alleviate public concerns.

Thirdly, our results demonstrate the positive moderating role of social connectedness in the relationship between identity salience and residents' engagement in place branding,

suggesting that local government and communities should facilitate social connectedness by building more effective communication social networks. The pandemic has changed individuals' social relationships and encouraged people to develop closer relationships with their neighbors. To ensure residents can gain sustained benefits from the collective community, local governments and communities should consider making an extra effort to develop interpersonal social relationships within the community. For example, digital and other mediated forms of communication can be used to create and nurture social connections.

*5.3. Study Limitations and Future Research Directions*

This study has several limitations that should be considered in further research. Firstly, to understand residents' reactions to a crisis related to COVID-19, we used data that were cross-sectional in nature. However, residents' reactions to the pandemic are dynamic. Therefore, further research may be needed to collect additional longitudinal data and, thus, add greater strength to the findings regarding the effect of residents' identity salience on their engagement in place branding. Secondly, this study used an online self-administered survey, which may have been subject to self-response bias. Further work could employ field studies to focus on residents' real-world place branding behavior. Thirdly, this study was set in China, and studies suggest that different cultural backgrounds can lead to different reactions to crises [105]. China is a collectivist society, where people pay more attention to interconnection and interdependence [99]. Therefore, it would be useful to conduct further studies in different cultural contexts.

**Author Contributions:** Conceptualization, W.H.; Validation, J.W.; Resources, J.W.; Data curation, Y.T.; Writing—original draft, W.H. All authors have read and agreed to the published version of the manuscript.

**Funding:** This research was supported by the Hainan Provincial Natural Science Foundation of China (721RC527, 721QN224).

**Informed Consent Statement:** Informed consent was obtained from all subjects involved in the study.

**Data Availability Statement:** Data are available on request.

**Conflicts of Interest:** The authors declare no conflict of interest.

**Appendix A**

The survey questionnaire

Thank you for participating in this questionnaire survey. This research aims to examine residents' engagement with place branding. This questionnaire will take about 15 min for you to complete. All data will be used for research purposes only and your responses will be kept strictly confidential.

---

**Personal information**

**1. Gender**: Male__       Female__

**2. Age**: 18–30__           31–40 __           41–50 __       51+__

**3. Monthly household income**:

Less than RMB5,000__       RMB5000-RMB1,5000__       More than RMB1,5000__

**4. Highest education attained**: Middle school or lower__          High school/vocational school __

College/University__               Master's or Ph. D.__

**5. Employed by the tourism sector**

Yes __               No __

---

| Identity Salience | Totally Agree | | | | | | Totally Disagree |
|---|---|---|---|---|---|---|---|
| 1. Sanya is a significant part of me. | 7 | 6 | 5 | 4 | 3 | 2 | 1 |
| 2. For me, Sanya means more than where I am living. | 7 | 6 | 5 | 4 | 3 | 2 | 1 |
| 3. The image of Sanya is something I usually think about. | 7 | 6 | 5 | 4 | 3 | 2 | 1 |
| **Residents' engagement in place branding (RE)** | | | | | | | |
| I contribute to improving residents' experience in Sanya. | 7 | 6 | 5 | 4 | 3 | 2 | 1 |
| I make efforts to strengthen the image of Sanya as a tourism destination. | 7 | 6 | 5 | 4 | 3 | 2 | 1 |
| I "talk up" Sanya to people I know. | 7 | 6 | 5 | 4 | 3 | 2 | 1 |
| **Community cohesion (CC)** | | | | | | | |
| I am closely related to my community. | 7 | 6 | 5 | 4 | 3 | 2 | 1 |
| I have developed fulfilling relationships with other members of the community. | 7 | 6 | 5 | 4 | 3 | 2 | 1 |
| I am always willing to help others in the community. | 7 | 6 | 5 | 4 | 3 | 2 | 1 |
| **Psychological empowerment (PE)** | | | | | | | |
| Tourism in Sanya makes me proud to be a Sanya resident. | 7 | 6 | 5 | 4 | 3 | 2 | 1 |
| Tourism in Sanya makes me feel special because my city is famous. | 7 | 6 | 5 | 4 | 3 | 2 | 1 |
| Tourism in Sanya makes me want to talk more about my city. | 7 | 6 | 5 | 4 | 3 | 2 | 1 |
| **Satisfaction with government (SWG)** | | | | | | | |
| The government made the right tourism decisions when the pandemic broke out. | 7 | 6 | 5 | 4 | 3 | 2 | 1 |
| The government did what was right in tourism when the pandemic broke out. | 7 | 6 | 5 | 4 | 3 | 2 | 1 |
| The government looked after the community's interests during the pandemic. | 7 | 6 | 5 | 4 | 3 | 2 | 1 |
| **Perceived economic benefit (PEB)** | | | | | | | |
| My family income is tied to tourism. | 7 | 6 | 5 | 4 | 3 | 2 | 1 |
| My income has decreased sharply because of the Sanya tourism industry's suffering. | | | | | | | |
| My economic future depends on tourism. | 7 | 6 | 5 | 4 | 3 | 2 | 1 |
| **Perceived risks (PR)** | | | | | | | |
| Tourists make me feel anxious during the pandemic period. | 7 | 6 | 5 | 4 | 3 | 2 | 1 |
| Tourists increase the risk of COVID-19 infection. | 7 | 6 | 5 | 4 | 3 | 2 | 1 |
| Tourists increase inconveniences in my daily activities. | | | | | | | |
| **Perceived social connectedness (SC)** | | | | | | | |
| I feel closely connected with my community during the pandemic. | 7 | 6 | 5 | 4 | 3 | 2 | 1 |
| I feel positive toward people connected with me during the pandemic. | 7 | 6 | 5 | 4 | 3 | 2 | 1 |
| Connection with others increases my belongingness during the pandemic. | 7 | 6 | 5 | 4 | 3 | 2 | 1 |

**Appendix B**

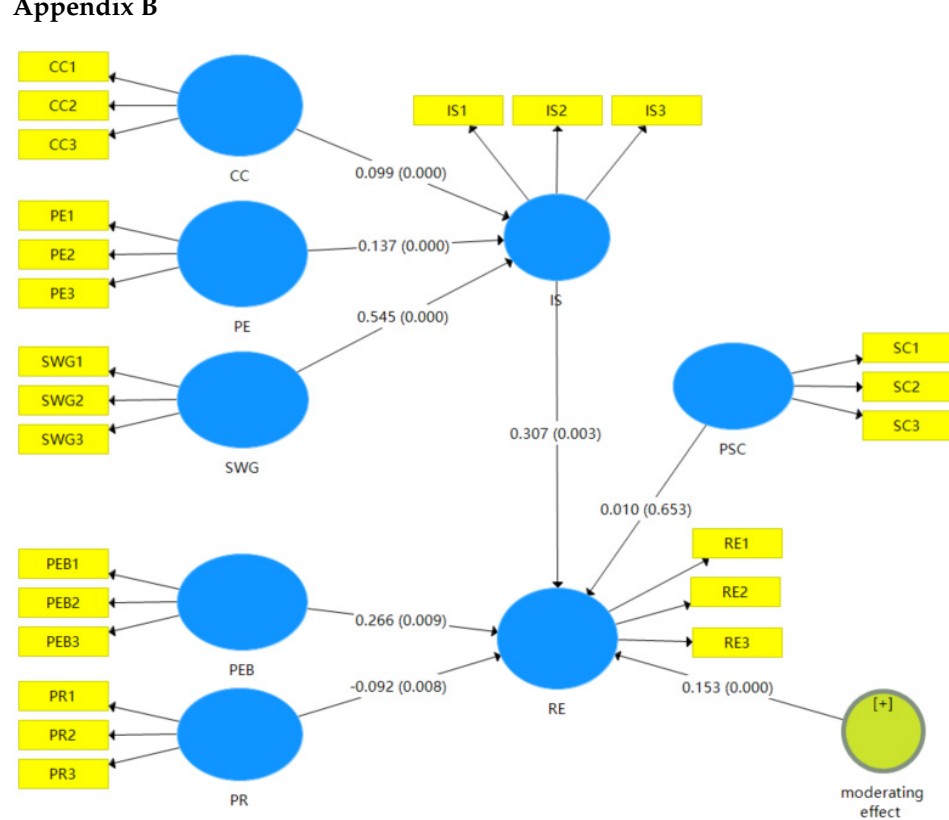

**Figure A1.** Structural model.

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
