# Peer review of "The Effect of Identity Salience on Residents’ Engagement with Place Branding during and Post COVID-19 Pandemic"

_sustainability, doi:10.3390/su15010357_

Round 1

Reviewer 1 Report

The role of the local community in the development of tourism is essential. In a period of crisis, the impact can be positive or negative, the study of this topic is important for local tourism organizations, and not only.

The paper is very well written and structured, the hypotheses are clearly argued, the references are rich and up to date.

The research method is adequate, and the results are clearly presented.

I recommend changing the RMB in USD or EUR to better understand the presented values (not all readers know automatically the currency parity).

Author Response

I recommend changing the RMB in USD or EUR to better understand the presented values (not all readers know automatically the currency parity).

Thank you for pointing this out. We have changed the RMB in USD in the revised vision.

Reviewer 2 Report

Dear Authors,

Thank you for the chance of reviewing your study aimed at investigating if the COVID-19 pandemic, as an environmental stimulus, could make residents’ collective identity salient, thus increasing their involvement and participation in destination branding activities. I find your application of the identity salience theory in the tourism field original and your framework effective to develop theoretical and practical implications. Overall, the quality of structure and quality is high. That said, I would suggest to pay attention to the following minor issues:

1.     Line 46: Check typo

2. Line 136-139: It is not clear where community cohesion, psychological empowerment, and satisfaction (as relationship-inducing factors) and economic benefits and perceived risk (as non-relationship-inducing factors) come from.  

Author Response

  1. Line 46: Check typo

Thank you for pointing this out. We had corrected this error in the revised version.

  1. Line 136-139: It is not clear where community cohesion, psychological empowerment, and satisfaction (as relationship-inducing factors) and economic benefits and perceived risk (as non-relationship-inducing factors) come from.

Thank you for your comment. The main purpose of this study was to explore the effect of the non-relationship inducing factors and non-relationship inducing factors on residents’ identity salience and engagement with place branding, so we developed hypotheses to test the effects of these factors. To make it more explicit, we had added explanations regarding why we choose these variables in the revised vision. It is shown in line 140-149:

Specifically, in this study, we use community cohesion, psychological empowerment and satisfaction with government as three relationship-inducing factors because they are critical factors that can facilitate identity related social relationship exchange and social interactions during and post the COVID-19 pandemic (Li et al., 2022; Philpot et al., 2021; Tedeschi & Calhoun, 2004; Woosnan et al., 2022). In contrast, we include economic benefits and perceived risk as two non-relationship-inducing factors because they can reflect the trade-off faced by the residents in a post-pandemic era, and both have significant effects on residents’ tourism related attitude and behavior (Li et al., 2022; Woosnam et al., 2022; Armutlu et al., 2021; Tilaki et al., 2021; Joo et al., 2021).

Reviewer 3 Report

The paper focuses on a very interesting theme. The goal of the paper is clear and is well motivated. The authors investigate the role of relationship-inducing and non-relationship-inducing factors in activating residents’ collective identity and engagement in destination branding. The study has the potential to make a useful contribution to the research areas of social commerce but the paper, in its current state, has several issues that need to be addressed. I have listed my concerns below.

1.In Literature Review sections, the authors can provide sufficient literature of satisfaction with the government, to support the investigation, and the logic relationships between satisfaction with the government and identity salience need to be strengthened.

2. The research design and methods section need to justify how the measurement items are developed and what are the basis for their selection and needs more detailed description for the items. What methods were used to control for the effects of possible confounding variables in order to improve the study's internal validity?

The measurement items can be provided in the appendix.

3.There is a mistake in line 45.

4. As acknowledged by the authors in the Limitations part, it’s a pity that the paper used an online self-administered survey instead of field study to investigate residents’ real-world place branding behavior.

Author Response

     1.In Literature Review sections, the authors can provide sufficient literature of satisfaction with the government, to support the investigation, and the logic relationships between satisfaction with the government and identity salience need to be strengthened.

Thank you for your comment. We had revised to make the logic clearer. Following your suggestion, we added two articles to support the relationship between satisfaction and identity salience. Besides, we added three new citations to support the significance of satisfaction with government on residents’ attitude and behavior toward tourism in the context of COVID-19, to make our argument more convincing. Please see line 204 to 229 in the revised vision:

According to Arnett et al.’s (2013) identity salience model, satisfaction is an important antecedent of identity salience. Additionally, the significance of satisfaction on identity salience has been verified by several studies in the organizational context. For example, Welborne and Cable (1995) showed satisfaction with the organization helped people reevaluate the salience of different identities. McCall and Simmons (1978) argue satisfaction with organizational events was significant for the development and maintenance of employees’ identities. In the tourism field, as an indicator of residence-place relationship, satisfaction with the local government has a direct impact on residents’ social identity and identification with the place (Chen & Dwyer, 2018, Chen et al., 2018). Studies have revealed that satisfaction with the local government is positively related to residents’ perceived positive impact of local community and tourism activities. For example, Wassler et al. (2021) found that trust in the government impacted residents’ identification with the place and ambassadorial behavior. Zuo et al. (2017) reported that mainland Chinese residents showed a higher level of support for “red tourism” if they were satisfied with the local government.

During the COVID-19 pandemic, the development of the travel industry has been highly contingent on local government travel restrictions (Koh, 2020), meaning residents’ attitudes and behavior toward tourism have been directly impacted by government policies (Woosnan et al., 2022). The significance of government in tourism post-pandemic recovery has been revealed by several studies. For example, Park et al. (2021) found residents satisfaction with government COVID-19 prevention measures significantly impacted their attitude toward tourism. Wong and Lai (2021) argue during the COVID-19 pandemic, satisfaction with government can increase residents’ confidence and support for tourism recovery. Furthermore, Fakfare and Sangpikul (2022) found residents satisfaction with government impacted their perception on tourism reactivation in a post pandemic era.

  1. The research design and methods section need to justify how the measurement items are developed and what are the basis for their selection and needs more detailed description for the items. What methods were used to control for the effects of possible confounding variables in order to improve the study's internal validity? The measurement items can be provided in the appendix.

Thank you for this comment. In the revised vision, we explained all the items we adopted from previous studies demonstrating high reliability, and suitable to be used in a time of COVID-19. Then, we conducted pretest to eliminate items with low communality or cross-loading of more than one factor. We provided our measurement items by appending our questionnaire. Please see the revision in line 356-357, 359-361:

These measurement items were adopted from previous studies demonstrating high re-liability and validity, and suitable to be used in a time of COVID-19. All items These 35 items were measured using a seven-point Likert scale ranging from 1 = totally disagree to 7 = totally agree. Then, we conducted a pretest and 11 items were eliminated due to low communality or cross-loading of more than one factor. Finally, a questionnaire with 24 measurement items were generated (appendix A).

and line 370-373:

Fourthly, we used control variables, including gender, age, household income and working industry, to ensure the observed effects are attribute to the core constructs. This helps control the effects of possible confounders, thereby improving internal validity (Woodruff, 2017).

      3.There is a mistake in line 45.

 Thank you for pointing this out. We had corrected this mistake in the revised version.

  1. As acknowledged by the authors in the Limitations part, it’s a pity that the paper used an online self-administered survey instead of field study to investigate residents’ real-world place branding behavior.

Thank you for this comment. At the time of this study, we had to use online self-administered survey because there was strict travel ban. We plan to investigate residents’ real-world behavior by filed study in our following studies.